# Permeation of AuCl$_4^-$ Across a Liquid Membrane Impregnated with A324H$^+$Cl$^-$ Ionic Liquid

**Francisco José Alguacil** and **Félix A. López** *

Department of Primary Metallurgy and Materials Recycling, National Center for Metallurgical Research, (CENIM-CSIC), Avda, Gregorio del Amo 8, 28040 Madrid, Spain; fjalgua@cenim.csic.es
* Correspondence: f.lopez@csic.es; Tel.: +34-915-538-900

**Abstract:** In the system Au(III)-HCl-A324H$^+$Cl$^-$, liquid-liquid extraction experiments were used to define the extraction equilibrium and the corresponding extraction constant; furthermore, the facilitated transport of this precious metal from HCl solutions across a flat-sheet supported liquid membrane was investigated using the same ionic liquid as a carrier, and as a function of different variables: hydrodynamic conditions, concentration of gold(III) (0.01–0.1 g/L), and HCl (0.5–6 M) in the feed phase, and carrier concentration (0.023–0.92 M) in the membrane. An uphill transport equation was derived considering aqueous feed boundary layer diffusion and membrane diffusion as controlling steps. The aqueous diffusional resistance ($\Delta_f$) and the membrane diffusional resistance ($\Delta_m$) were estimated from the proposed equation with values of 241 s/cm and 9730 s/cm, respectively. The performance of the present carrier was compared against results yielded by other ionic liquids, and the influence that other metals had on gold(III) transport from both binary or quaternary solutions was also investigated. Gold was finally recovered from receiving solutions as zero valent gold nanoparticles.

**Keywords:** liquid membrane; ionic liquid; gold

## 1. Introduction

The continuous growth of residues generated in cities make these a source for the recovery of the valuable metals contained in them, and thus, nowadays, the concept of urban mining is taking more importance. Among these residues, the materials generated from the electronic and communication industries are of importance [1,2] because they contain, among other valuable metals, gold. Thus, there is an interest from worldwide researchers to investigate methods for the recovery of this precious metal from these residues or, in general, gold-bearing wastes [3–8].

This recovery can be executed by pyro or hydrometallurgical procedures, and in the case of the latter methodologies, and after the leaching step, a multi-elemental solution is generally obtained, and thus, the separation of gold (or the most valuable element) from the others is of paramount importance. In the separation technologies, liquid membranes are gaining major importance due to its operational characteristics, and the possibility of the treatment of dilute solutions more efficiently solvent extraction. These liquid membranes technologies include supported liquid membranes combined with the kinetic properties of membrane processes, with the chemical reaction and selectivity that the organic extractants, imbibed in the micropores of the support, provide to the system, thus resulting in the facilitated coupled transport of metal species. From the types of organic extractants, ionic liquids constitute a subcategory also of current interest, due to the special characteristics that these reagents present, which somewhat allows them to be referred to as green solvents. These ionic liquids are used in different fields [9–12] including the removal of metals from aqueous solutions [13–16].

The present work links all of the above-mentioned: (i) the recovery of gold(III) from acidic solutions; (ii) using a supported liquid membrane in a flat-sheet configuration, as a previous step to its scaling up to a hollow fiber module, and (iii) the use of an ionic liquid as a carrier in the membrane system.

As an ionic liquid, quaternary ammonium salt, derived from the reaction of a tertiary amine (Hostarex A324) with hydrochloric acid was used, and different variables affecting the metal transport were investigated. The results were also compared with those obtained with other ionic liquids, and the competitive transport of gold(III) against Cu(II), Fe(III), and Ni(II) as representative elements of those found in electronic wastes, were investigated from binary and quaternary solutions. As a final step, gold was recovered as zero valent nanogold by precipitation of the receiving solution with sodium borohydride.

## 2. Materials and Methods

The tertiary amine Hostarex A324 (Hoechst AG, actually part of Sanofi) has tri-isooctyl amine as its active component, and was used without further purification. It was dissolved in Solvesso 100 (aromatic diluent, Exxon Chem. Iberia, Madrid, Spain) to attain various objectives: (i) to decrease the high viscosity of the extractant (amine or the derived ionic liquid); and (ii) as the extractants are expensive, the use of a diluent allows for an adequate concentration of the carrier for specific purposes of the system to be investigated, avoiding the presence of an unused excess of the reagent in the operation. Other chemicals used in the work were analitic grade, except the ionic liquids Cyphos IL 101, Cyphos IL102 (phosphonium derivatives), and the precursor Primene JMT (aliphatic primary amine), which were obtained from Cytec (actually part of Solvay) and Rohm and Haas, respectively.

Liquid-liquid extraction experiments were carried out in thermostatted separatory funnels that were mechanically shaken. After the generation of the ionic liquid [17], organic solutions (25 mL) of various concentrations of the ionic liquid were put into contact with aqueous solutions (25 mL) of $7.1 \times 10^{-5}$ Au(III) in 1 M HCl and shaken for 15 min, enough time to achieve equilibrium at 20 °C. After phase disengagement of less than 3 min, the residual gold content in the raffinate was analyzed by atomic absorption spectrometry (Perkin Elmer 1100B, Waltham, MA, USA), and the gold concentration in the equilibrated organic phases was calculated by the mass balance.

Supported liquid membrane experiments were carried out using the same cell and procedure described elsewhere [18]. Millipore Durapore GVH4700, (polyvinylidenedifluoride, PVDF, material) was used to support the carrier phase. The characteristics of the support were $12.5 \times 10^{-3}$ thickness (dm), 75% porosity ($\varepsilon$), and 1.67 tortuosity ($\tau$).

Gold precipitation experiments were carried out at 20 °C in a glass reaction vessel under slight stirring conditions (100 rpm) and used solid sodium $NaBH_4$ as a reducing agent. Since the reduction reaction was almost instantaneous, no time measurements were necessary in this step. After filtration and washing with distilled water, the precipitate was dried and stored in a dessicator.

Gold (and metal) concentrations from both aqueous solutions, source and receiving, were analyzed from samples taken at elapsed times using the same AAS method as above, and the permeability coefficient P was estimated from Equation (1), derived from the combination of gold (and metal) balance in the source phase and Fick´s law in the membrane phase [19–21]:

$$\ln \frac{[\text{Au}]_t}{[\text{Au}]_0} = -\frac{AP}{V}t \tag{1}$$

where $[\text{Au}]_t$ and $[\text{Au}]_0$ are the gold (metal) concentrations in the source phase at the elapsed time and time zero, respectively; A is the membrane area (11.3 cm$^2$), V is the volume of the source phase (200 mL); P is the permeability coefficient, and t is the elapsed time.

## 3. Results

### 3.1. Liquid-Liquid Extraction

This series of experiments used organic phases of the ionic liquid (0.012–0.0012 M) in Solvesso 100 and an aqueous solution, the composition of which was given in Section 2. The results from these experiments were shown in Table 1.

**Table 1.** Liquid-liquid extraction of gold(III) from 1 M HCl solutions by the ionic liquid $A324H^+Cl^-$ dissolved in Solvesso 100.

| $[A324H^+Cl^-]$ (M) | $^aD_{Au}$ |
|---|---|
| 0.012 | 21.2 |
| 0.0058 | 7.3 |
| 0.0023 | 2.1 |
| 0.0012 | 1.4 |

a—The distribution coefficient $D_{Au}$ was calculated as the ratio of gold concentrations in the organic and aqueous phases at the equilibrium.

The results were treated by a tailored computer program that compared the experimental distribution coefficient ($D_{exp}$) results with the calculated distribution coefficients ($D_{cal}$) values, minimizing the expression:

$$U = \Sigma\left(\log D_{cal} - \log D_{exp}\right)^2 \tag{2}$$

The results from this fit showed that the extraction of gold(III) by the ionic liquid $A324H^+Cl^-$ dissolved in Solvesso 100 responded to an anion exchange equilibrium as:

$$AuCl_{4_{aq}}^- + A324H^+Cl_{org}^- \Leftrightarrow A324H^+AuCl_{4_{org}}^- + Cl_{aq}^- \tag{3}$$

with the equilibrium constant defined as:

$$K = \frac{\left[A324H^+AuCl_4^-\right]_{org}\left[Cl^-\right]_{aq}}{\left[AuCl_4^-\right]_{aq}\left[A324H^+Cl^-\right]_{org}} \tag{4}$$

and log K = 3.11 ± 0.18 and U = $4.5 \times 10^{-2}$. In the above equations, the subscripts aq and org represent the aqueous and organic phases, respectively.

### 3.2. Supported Liquid Membrane Transport

#### 3.2.1. Influence of the Stirring Speed

The influence of the stirring speed was investigated in order to optimize uniform mixing in the feed solution and to minimize the thickness of the aqueous feed boundary layer with the feed and receiving conditions being maintained as 0.01 g/L Au(III) in 1 M HCl and 0.1 M NaSCN, respectively. The extractant concentration was 0.23 M in Solvesso 100 immobilized on a Durapore microporous support. The permeability coefficient becomes virtually independent of the stirring speed from 500 rpm indicating first a decrease in the aqueous feed boundary layer thickness, and then a minimum value of the thickness was reached from 500 rpm (Table 2). The stirring speed of 750 rpm was kept constant throughout further experimentation.

**Table 2.** Dependence of gold(III) permeabilities values on the stirring speed of the feed phase.

| Stirring Speed (rpm) | $P \times 10^3$ (cm/s) |
|:---:|:---:|
| 375 | 2.1 |
| 425 | 3.0 |
| 500 | 3.9 |
| 750 | 3.9 |
| 1000 | 3.9 |

Stirring speed receiving phase: 500 rpm.

In the case of the receiving phase, and considering that the stirrer in the cell was very close to the membrane support, the thickness of the boundary layer was minimized; thus, the resistance in the receiving side can be neglected [22], and the stirring speed of 500 rpm was used in the receiving side throughout all the experiments.

### 3.2.2. Effect of Receiving Phase Composition on Permeability of Gold(III)

Sodium thiocyanate solutions were used as the receiving phase for gold(III) due to the ability of thiocyanate to form a stable complex with gold(III), $Au(SCN)_4{}^-$ with log $\beta_2$ of 42 [23]. The feed solution had the same composition than above, whereas the receiving solution was of 0.1, 0.25, and 0.5 M sodium thiocyanate. Results derived from this experimentation indicated that the increase in the sodium thiocyanate composition, up to 0.5 M, had a negligible effect on gold permeation.

### 3.2.3. Effect of HCl Concentration in the Feed Phase on Permeability of Gold(III)

The single gold transport across the supported liquid membrane (SLM) of 0.01 g/L Au(III) from the aqueous feed phases of varying (0.1–6 M) HCl concentrations was investigated using 0.1 M NaSCN as the receiving phase. Figure 1 shows that the time dependent fraction ln $[Au]_t/[Au]_0$ in the feed phase at the various HCl concentrations indicated a strong dependence on the permeability of gold with this variable. A maximum in permeability (Table 3) was obtained at 1 M HCl, whereas the progressive decrease in this value was attributable to one or both of the next factors: (i) increase in the ionic strength, and (ii) shift of the next equilibrium to the right, thus, decreasing the availability of $AuCl_4{}^-$ species with the increase in the HCl concentration:

$$AuCl_4^- + H^+ \Leftrightarrow HAuCl_4 \tag{5}$$

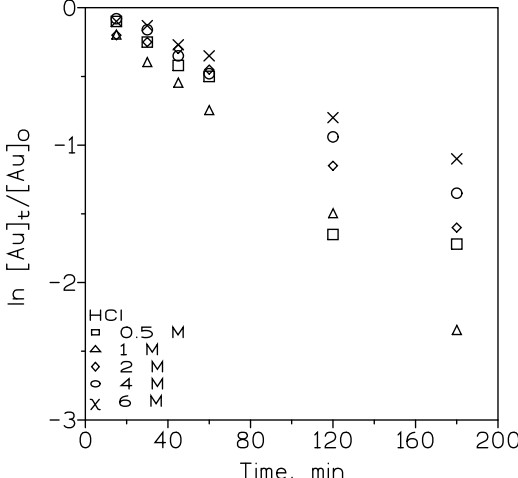

**Figure 1.** Influence of the HCl concentration on the permeability of gold(III) as a function of the ln $[Au]_t/[Au]_0$ feed phase: 0.01 g/L Au(III) in HCl. Membrane phase: 0.23 M ionic liquid in Solvesso 100 on Durapore support. Receiving phase: 0.1 M NaSCN.

**Table 3.** Gold(III) permeabilities at various HCl concentrations in the feed phase.

| HCl (M) | $P \times 10^3$ (cm/s) |
|---------|------------------------|
| 0.5 | 2.8 |
| 1 | 3.9 |
| 2 | 2.6 |
| 4 | 2.2 |
| 6 | 1.8 |

Experimental conditions, as in Figure 1.

### 3.2.4. Effect of Carrier Concentration on Permeability of Gold(III)

The results in relation with the transport of gold(III) from a feed phase containing 0.01 g/L Au(III) in 1 M HCl and the receiving solution 0.1 M NaSCN, and varying concentrations of the ionic carrier in the range 0.023–0.92 M dissolved in Solvesso 100 revealed an increase of the gold(III) permeability up to 0.17 M, then a maximum value of P was obtained in the 0.17–0.23 M range (Table 4). These results indicated that the transport process was first dominated by diffusion in the organic membrane, and then a limiting permeability value $P_{lim}$ was obtained, with this situation indicative of a transport process controlled by diffusion in the stagnant film of the feed phase. In the limiting situation:

$$P_{\text{lim}} = \frac{D_{aq}}{d_{aq}} \tag{6}$$

and $d_{aq}$ was estimated as $2.6 \times 10^{-3}$ cm, considering the average coefficient of species in the aqueous phase $D_{aq}$ as $10^{-5}$ cm²/s, and $P_{lim}$ of $3.9 \times 10^{-3}$ cm/s. This $d_{aq}$ represents the thickness of the feed boundary layer. The decrease in P value at higher carrier concentrations (0.46–0.92 M) in Solvesso 100 can be attributed to a gradual increase in the organic solution viscosity, with the increase in the carrier concentration, which augmented the membrane resistance to the transport.

**Table 4.** Variation of gold(III) permeation with the carrier concentration.

| Carrier Concentration (M) | $P \times 10^3$ (cm/s) |
|---------------------------|------------------------|
| 0.023 | 1.8 |
| 0.058 | 2.6 |
| 0.12 | 3.2 |
| 0.23 | 3.8 |
| 0.17 | 3.9 |
| 0.46 | 3.4 |
| 0.92 | 2.9 |

### 3.2.5. Influence of Metal Concentration on Permeability of Gold(III)

Examination of the effect of the initial concentration of gold(III) (0.01–0.1 g/L) in the feed phase; when the receiving phase contained no gold concentration, it was revealed that the initial metal flux:

$$J = [\text{Au}]_0 P \tag{7}$$

initially increased sharply from 0.01 g/L to 0.05 g/L and beyond this became almost independent of the initial concentration (0.05–0.1 g/L). The results of gold flux across the membrane as a function of gold(III) concentration are shown in Figure 2. The initial increase in gold flux is in accordance with the expected trend, as indicated by Equation (7), since the flux varied with metal concentration. The above rule was obeyed up to 0.05 g/L concentration of gold, beyond which the initial flux tends to be a constant value, attributed to a saturation of the membrane pores, which resulted in flux maximization, and the creation of a metal-carrier layer on the feed-membrane interface.

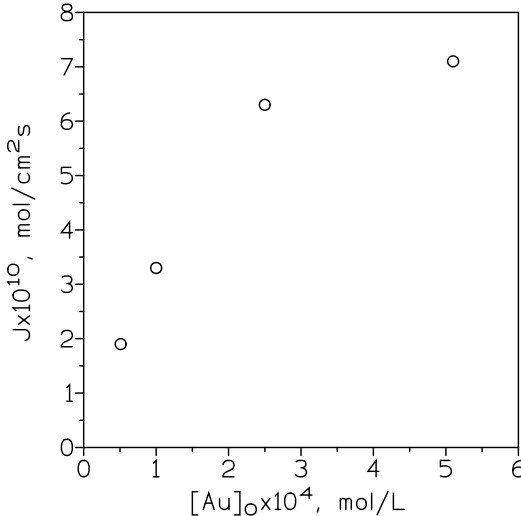

**Figure 2.** Influence of the initial gold(III) concentration on the initial flux. Feed phase: Gold(III) in 1 M HCl. Carrier phase:0.23 M ionic liquid in Solvesso 100 immobilized on Durapore support. Receiving phase: 0.1 M NaSCN.

### 3.2.6. Influence of the Ionic Strength of the Feed Phase on the Permeation of Gold(III)

This effect was investigated by varying the source of $Cl^-$ ions in the feed phase with other experimental conditions as follows: feed phase of 0.01 g/L Au(III) in 1 M (H,Li,Na)Cl, membrane phase of 0.23 M ionic liquid in Solvesso 100, and receiving phase of 0.1 M sodium thiocyanate. From the results presented in Table 5, it can be seen that the substitution of HCl for LiCl or NaCl resulted in a decrease of gold(III) permeation, whereas the influence of changing LiCl by NaCl had a negligible effect on Au(III) transport. This variation can be attributed to the different activity of $H^+$ ions in the solution with respect to $Li^+$ and $Na^+$, and that the driven force for gold(III) transport was the different acidity of the feed and the receiving solutions.

**Table 5.** Gold(III) transport varying the chloride source in the feed solution.

| System | $P \times 10^3$ (cm/s) |
|---|---|
| 1 M HCl | 3.9 |
| 1 M LiCl | 2.3 |
| 1 M NaCl | 2.5 |

Support: Durapore GVHP4700.

### 3.2.7. Estimation of Diffusional Parameters

Figure 3 shows the gold(III) concentration profile in the three phases of the system.

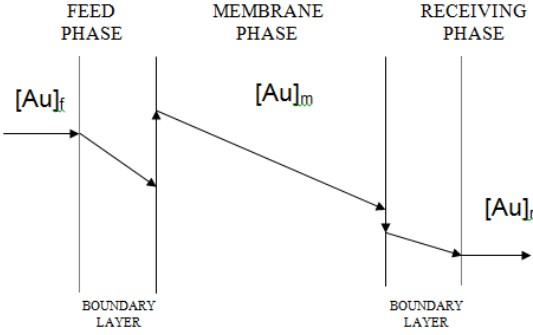

**Figure 3.** Gold profile in the supported liquid membrane (SLM)

Taking into account the extraction equilibrium and constant represented in Equations (2) and (3), respectively, and that the concentration of the AuCl$_4$$^-$ carrier complex in the membrane phase at the receiving side may be negligible compared to that at the feed side, a final expression for the permeability coefficient can be written as:

$$P = \frac{K[carrier][Cl^-]^{-1}}{\Delta_m + \Delta_f K[carrier][Cl^-]^{-1}} \tag{8}$$

and,

$$\frac{1}{P} = \Delta_f + \frac{\Delta_m}{K[carrier][Cl^-]^{-1}} \tag{9}$$

The above expressions combined in one equation the diffusion and equilibrium parameters involved in the Au(III) transport. In Equations (8) and (9), $\Delta_f$ and $\Delta_m$ are the transport resistances due to diffusion in the feed and membrane, respectively, and K is the extraction equilibrium constant.

In a plot of 1/P versus 1/K[carrier][Cl$^-$]$^{-1}$ for different carrier concentrations and a chloride ion concentration of 1 M, a straight line with slope $\Delta_m$ (9730 s/cm) and ordinate $\Delta_f$ (241 s/cm) was obtained (Figure 4).

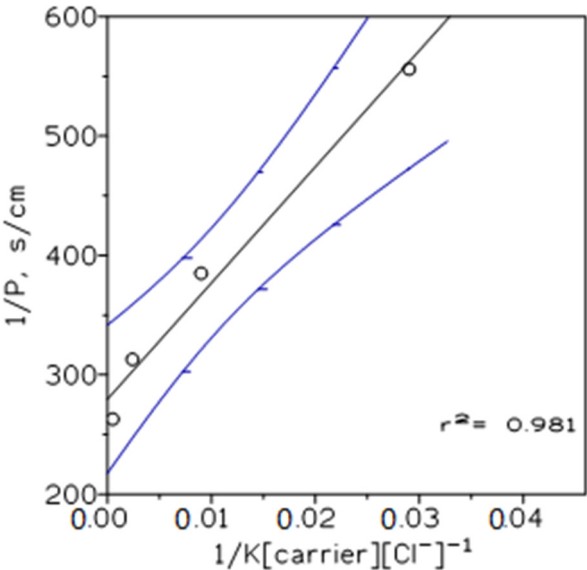

**Figure 4.** Plot of 1/P versus 1/K[carrier][Cl$^-$]$^{-1}$. Dotted line indicates a 95% confidence interval of the regression line.

The estimated value of the membrane diffusion coefficient:

$$D_m = \frac{d_m}{\Delta_m} \tag{10}$$

was calculated as $1.3 \times 10^{-6}$ cm$^2$/s. The diffusion coefficient of the gold-carrier complex in the bulk of the membrane phase can be estimated by:

$$D_{m,b} = D_m \frac{\tau^2}{\varepsilon} \tag{11}$$

and D$_{m,b}$ as $4.8 \times 10^{-6}$ cm$^2$/s. It can be seen that the value of D$_m$ was lower than the value of D$_{m,b}$, which can be attributed to the diffusional resistance caused by the microporous thin support separating the feed and receiving phases. The mass transfer coefficient in the feed phase, $\Delta_f$$^{-1}$ was estimated to be $4.1 \times 10^{-3}$ cm/s.

3.2.8. Gold(III) Permeation Using Different Ionic Liquids as Carriers for Gold Transport

Several ionic liquids (Table 6) were used as carriers to compare their performance in gold(III) transport with that of $A324H^+Cl^-$.

**Table 6.** Ionic liquids used in the transport of gold(III).

| Name and Acronym | Active Group |
|---|---|
| Cyphos IL101 | quaternary phosphonium chloride salt [a] |
| Cyphos IL 102 | quaternary phosphonium bromide salt [a] |
| Aliquat 336 | quaternary ammonium chloride salt [b] |
| Primene JMT, $PJMTH^+Cl^-$ | quaternary ammonium chloride salt [c] |

a—Same organic radicals, the only difference is the anion. b—Ammonium group derived from a tertiary amine. c—Ammonium group derived from a primary amine.

In this series of experiments, the feed solution contained 0.01 g/L Au(III) in 1 M HCl, and the membrane phase was 0.17 M of the corresponding ionic liquid in Solvesso 100 supported on a Durapore GVHP4700 support; as the receiving phase, a 0.1 M sodium thiocyanate was used throughout the tests. Results from this investigation are shown in Figure 5.

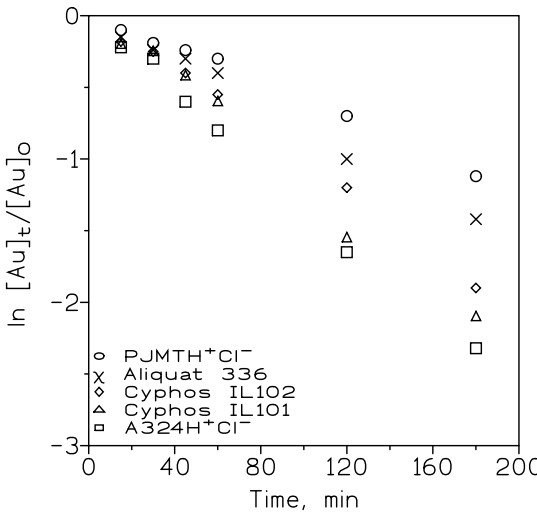

**Figure 5.** Permeability behavior of gold(III) using various ionic liquids as carriers.

Practically the same permeabilities coefficients were obtained when $A324H^+Cl^-$ ($3.8 \times 10^{-3}$ cm/s) and Cyphos IL101 ($3.4 \times 10^{-3}$ cm/s) were used as carriers for gold(III) transport, and then the sequence was Cyphos IL102 > Aliquat 336 > $PJMTH^+Cl^-$, with permeability coefficients of $3.1 \times 10^{-3}$, $2.3 \times 10^{-3}$, and $1.8 \times 10^{-3}$ cm/s, respectively.

3.2.9. Gold(III) Permeation from Multi-Elemental Feed Solutions

The permeation of gold(III) in the presence of other metal ions, generally accompanying the precious metal, was investigated using binary and quaternary-metal solutions. In the case of the binary solutions, these contained gold(III) in an equimolar concentration ($5.1 \times 10^{-3}$ M) with that of the corresponding ion (Cu(II), Fe(III) or Ni(II)) in 1 M HCl, the organic solutions were of 0.23 M carrier in Solvesso 100 immobilized in the support, and the receiving solution was of 0.1 M NaSCN. The results from these set of experiments, in the form of the separation factors, are shown in Table 6. The separation factors are calculated as:

$$\beta_{Au/M} = \frac{P_{Au}}{P_M} \tag{12}$$

Thus, this carrier offered good selectivity for the transport of gold(III) from these metals. This investigation were also done using a quaternary metal-bearing solution, and with the experimental conditions fixed as above. Figure 6 presents the results of Au(III) and metal transport, where the clean permeation of gold(III) was observed, and thus, the selectivity of this ionic liquid with respect to gold(III) (Table 7), and an apparent permeation order Au(III) > Fe(III) > Cu(II) >> Ni(II) was established under the experimental conditions used in this work. It is worth mentioning here that the presence of these metal ions in the feed phase did not appreciably influence the permeation of gold(III) with respect to what was obtained when the feed solution only contained the precious metal.

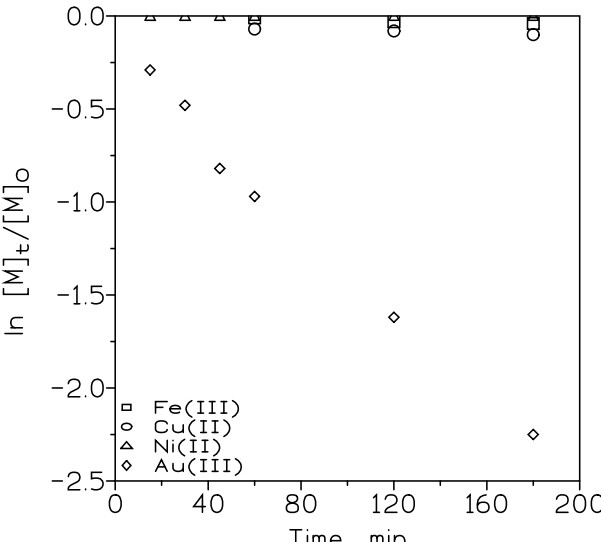

**Figure 6.** Transport of gold(III) in the presence of Fe(III), Cu(II), and Ni(II).

**Table 7.** Separation factors for the transport of gold(III) in the presence of other metals.

| Feed Phase | $\beta_{Au/M}$ |
|---|---|
| Au(III)-Cu(II) | 19 |
| Au(III)-Fe(III) | 31 |
| Au(III)-Ni(II) | nickel was not transported |
| Au(III)-Cu(II)-Fe(III)-Ni(II) | 23 (Cu(II)), 57 (Fe(III)), no transport of Ni(II) |

### 3.2.10. Precipitation of Nanogold Particles

Once gold(III) was released to the receiving phase, it was recovered from it by precipitation as zero valent gold nanoparticles with sodium borohydride. This salt produced hydrogen when added to the aqueous solution:

$$BH_4^- + 4H_2O \rightarrow B(OH)_4^- + 4H_2 \tag{13}$$

and this hydrogen reacted with the gold(III)-thiocyanate complex to produce zero valent gold:

$$2Au(SCN)_4^- + 3H_2 \rightarrow 2Au^0 + 6H^+ + 8SCN^- \tag{14}$$

This zero valent gold was released as gold nanoparticles (Figure 7).

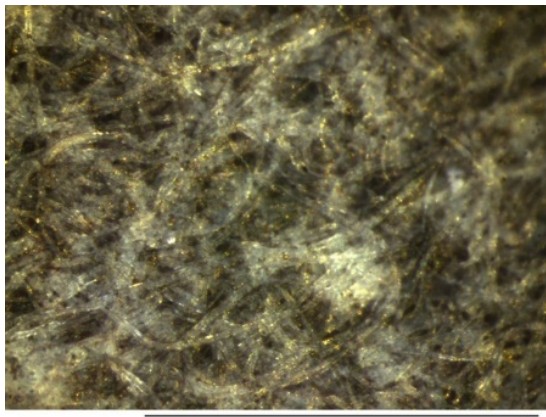

**Figure 7.** Gold nanoparticles after precipitation with sodium borohydride of the gold(III)-thiocyanate solution.

## 4. Conclusions

It was estimated that the value of the extraction equilibrium constant was $1.3 \times 10^3$ for gold(III) extraction from 1 M HCl solutions with the ionic liquid A324H$^+$Cl$^-$ dissolved in Solvesso 100. This ionic liquid effectively transported gold(III) from the HCl solutions, however, a maximum in gold(III) permeation was obtained at a HCl concentration of 1 M in the feed phase. At low carrier concentrations, gold transport is controlled by the diffusion across the membrane, while at medium carrier concentrations, which corresponded to the limiting conditions (Equation (6)). The transport process is controlled by diffusion in the feed phase boundary layer, and finally at very high carrier concentrations in the membrane phase, gold transport decreases due to an increase in the membrane resistance resulting from the increase of the viscosity of the carrier solution imbibed in it. An equation, which included both diffusional and equilibrium parameters, was derived, allowing for the estimation of mass transfer coefficients as $4.1 \times 10^{-3}$ cm/s and $1.0 \times 10^{-4}$ cm/s for the feed and the membrane phases, respectively. Au(III) can be separated from Cu(II), Fe(III), or Ni(II), and gold is recovered from the receiving solution as zero valent gold nanoparticles.

**Author Contributions:** Conceptualization, F.J.A.; Methodology, F.J.A. and F.A.L.; Formal analysis, F.J.A. and F.A.L.; Investigation, F.J.A.; Resources, F.A.L.; Writing—original draft preparation, F.J.A.; Writing—review and editing, F.J.A. and F.A.L. All authors have read and agreed to the published version of the manuscript.

**Funding:** This research received no external funding.

**Acknowledgments:** We acknowledge the support of the publication fee by the CSIC Open Access Publication Support Initiative through its Unit of Information Resources for Research (URICI).

**Conflicts of Interest:** The authors declare no conflicts of interest.

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
