# Peer review of "Permeation of AuCl4 Across a Liquid Membrane Impregnated with A324H+Cl Ionic Liquid"

_metals, doi:10.3390/met10030363_

Round 1

Reviewer 1 Report

Dear author!

Your article "Permeation of AuCl4- across a liquid membrane impregnated with A324H+Cl- ionic liquid" is of great interest to the scientific community.

I recommend the article for publication after minor corrections. I have a few questions and comments:

  1. During the experiments the initial concentrations of all metals (Au(III), Ni(II), Fe(III), Cu(II)) are equal. However, in real e-waste leaching solutions the concentration of accompanying metals is several orders of magnitude higher than that of gold (III). Thus, even a slight transition of other metal ions in absolute values will be comparable to the amount of gold recovered. You should take this fact into account.
  2. Often, along with noble and non-ferrous metals, REMs are present in electronic waste. Is the proposed method of gold extraction suitable if such metals are present in the solution?
  3. It is necessary to explain why, unlike gold (III), the remaining metals do not pass into the receiving phase through the membrane.
  4. Some sentences need to be reformulated (e.g. lines 9-14; 341-346), as it look too cumbersome. This makes it difficult to read and understand the text.
  5. For all reagents used, it must be specified: (organization name, city, country)
  6. The Materials and Methods section does not specify the method and equipment with which Figure 7 was obtained.

Author Response

Reviewer 1.

Your article "Permeation of AuCl4- across a liquid membrane impregnated with A324H+Cl- ionic liquid" is of great interest to the scientific community.

I recommend the article for publication after minor corrections. I have a few questions and comments:

Thank you for your above comments.

During the experiments the initial concentrations of all metals (Au(III), Ni(II), Fe(III), Cu(II)) are equal. However, in real e-waste leaching solutions the concentration of accompanying metals is several orders of magnitude higher than that of gold (III). Thus, even a slight transition of other metal ions in absolute values will be comparable to the amount of gold recovered. You should take this fact into account.

Yes you are right and we agree with you. In our manuscript, and in order to gain knowledge about the co-transport (or not) of other elements and their influence (or not) on gold(III) transport, we use equimolar concentrations. The fact that you indicate may be true, what we think that it can be mitigate by the use of a more efficient operation, i.e. hollow fiber module, and/or lowering the carrier concentration to limits in which the gold(III) transport will be good enough and the co-transport of the accompanying metals (even when present in the feed solution at higher concentrations) will be negligible.   

Often, along with noble and non-ferrous metals, REMs are present in electronic waste. Is the proposed method of gold extraction suitable if such metals are present in the solution?

It is dependent of the form in which REMs will be in the feed solution, if they are present in cationic form, there will be no transport and gold(III) separation will be quantitative, if REMs form any type of anionic complexes (i.e. chloride), experimentation will be necessary to show what happens, at the present time we can not give any more data about this issue.

It is necessary to explain why, unlike gold (III), the remaining metals do not pass into the receiving phase through the membrane.

We give a brief explanation on the revised text, basically this behaviour responds to a chemical affinity of the corresponding metal-chloride complex with the ionic liquid, and  to the interchange the chloride ion, to the kinetics characteristics accompanying the transport of any metal-complex, and in conclusion to the permeation order characteristic of any system (which must be determined experimentally) and it is dependent of the whole experimental characteristics used in the given system (even the diluent used in the organic phase).

Some sentences need to be reformulated (e.g. lines 9-14; 341-346), as it look too cumbersome. This makes it difficult to read and understand the text.

We do it.

For all reagents used, it must be specified: (organization name, city, country)

Please note that we use the Journal Editorial style for this.

The Materials and Methods section does not specify the method and equipment with which Figure 7 was obtained.

We include it now.

Reviewer 2 Report

The manuscript by Francisco Jose Alguacil and Felix A. Lopez reports experimental study of liquid-liquid extraction to define the extraction equilibrium and the corresponding extraction constant. The problem is actual, and its solution is demanded by practical necessity. Authors combined three known approaches and received appropriate new results, which can be interesting for corresponding applications.

The publication of the manuscript isn’t able not only to bring essential harm, but also can provide some new contribution to the existing amount of the information on this subject. The language is well enough to make understandable all steps of the investigation. Result appears rather similar to quite reliable one.

So, the manuscript might be recommended for the publication.

Author Response

The manuscript by Francisco Jose Alguacil and Felix A. Lopez reports experimental study of liquid-liquid extraction to define the extraction equilibrium and the corresponding extraction constant. The problem is actual, and its solution is demanded by practical necessity. Authors combined three known approaches and received appropriate new results, which can be interesting for corresponding applications.

The publication of the manuscript isn’t able not only to bring essential harm, but also can provide some new contribution to the existing amount of the information on this subject. The language is well enough to make understandable all steps of the investigation. Result appears rather similar to quite reliable one.

So, the manuscript might be recommended for the publication.

Thank you for your positive comments.

Reviewer 3 Report

Authors report a methodology based on liquid-liquid experiments that can be used to extract Au(III) from aqueous solutions

The procedure is interesting and can represent a valid Au(III) exstraction method.

However, some revision are needed:

  • section 3.1 needs to be more explicit. For example in line 104 authors said "it was shown…". How was it shown? the authors must explain better.
  • section 3.2.6. Authors analyse the effect of the ionic strength on permeation, but they always used 1:1 electrolytes (HCl, NaCl or LiCl) 1 M. In this way, ionic strength is always the same and change the ionic medium only. Authors must reconsider this section in this perspective
  • along all the manuscript, errors (for example standard deviation) on experimental data are never reported. they must absolutely be added

Minor revision:

check the upper and subscript along all manuscript (e.g. lines 338, 339, 347)

Author Response

Authors report a methodology based on liquid-liquid experiments that can be used to extract Au(III) from aqueous solutions

The procedure is interesting and can represent a valid Au(III) extraction method.

Thank you for your comments.

However, some revision is needed:

Section 3.1 needs to be more explicit. For example in line 104 authors said: "it was shown…". How was it shown? The authors must explain better.

It was corrected the sentence.

Section 3.2.6. Authors analyze the effect of the ionic strength on permeation, but they always used 1:1 electrolytes (HCl, NaCl or LiCl) 1 M. In this way, ionic strength is always the same and change the ionic medium only. The authors must reconsider this section from this perspective.

In this work, we only use chloride salts, in order to avoid the introduction, in the system, of another anion different to chloride, which more or less responded to a real leaching situation of electronic or scrap jewelry materials with aqua regia.  

Along all the manuscript, errors (for example standard deviation) on experimental data are never reported. They must absolutely be added.

But please note that this type of information was not the norm in this type of publication including papers on liquid-liquid extraction, adsorption, ion exchange resins. We think that the type of information that you requested was more for an Analytical Chemistry manuscript, which evidently the present one was not.   

Minor revision:

check the upper and subscript along all manuscript (e.g. lines 338, 339, 347)

We do it.

Round 2

Reviewer 3 Report

Authors replyed clearly to all my comments.

In my opinion, manuscript can be accepted in this form